# MicroRNA Expression Profiles in Human Samples and Cell Lines Revealed Nine miRNAs Associated with Cisplatin Resistance in High-Grade Serous Ovarian Cancer

**DOI:** 10.3390/ijms25073793

**Published:** 2024-03-28

**Authors:** Marienid Flores-Colón, Mariela Rivera-Serrano, Víctor G. Reyes-Burgos, José G. Rolón, Josué Pérez-Santiago, María J. Marcos-Martínez, Fatima Valiyeva, Pablo E. Vivas-Mejía

**Affiliations:** 1Department of Biochemistry, University of Puerto Rico, Medical Sciences Campus, San Juan, PR 00936, USA; marienid.flores@upr.edu (M.F.-C.); victor.reyes@upr.edu (V.G.R.-B.); 2Comprehensive Cancer Center, University of Puerto Rico, San Juan, PR 00936, USA; mariela.rivera20@upr.edu (M.R.-S.); joperez@cccupr.org (J.P.-S.); fvaliyeva@cccupr.org (F.V.); 3Department of Biology, University of Puerto Rico, Rio Piedras Campus, San Juan, PR 00936, USA; 4School of Medicine, University of Puerto Rico, Medical Sciences Campus, San Juan, PR 00936, USA; jose.rolon6@upr.edu; 5Department of Pathology and Laboratory Medicine, University of Puerto Rico, Medical Sciences Campus, San Juan, PR 00936, USA; maria.marcos@upr.edu

**Keywords:** high-grade serous ovarian cancer, ovarian cancer, miRNAs, cisplatin, miRNA inhibitors

## Abstract

Metastasis and drug resistance are major contributors to cancer-related fatalities worldwide. In ovarian cancer (OC), a staggering 70% develop resistance to the front-line therapy, cisplatin. Despite proposed mechanisms, the molecular events driving cisplatin resistance remain unclear. Dysregulated microRNAs (miRNAs) play a role in OC initiation, progression, and chemoresistance, yet few studies have compared miRNA expression in OC samples and cell lines. This study aimed to identify key miRNAs involved in the cisplatin resistance of high-grade-serous-ovarian-cancer (HGSOC), the most common gynecological malignancy. MiRNA expression profiles were conducted on RNA isolated from formalin-fixed-paraffin-embedded human ovarian tumor samples and HGSOC cell lines. Nine miRNAs were identified in both sample types. Targeting these with oligonucleotide miRNA inhibitors (OMIs) reduced proliferation by more than 50% for miR-203a, miR-96-5p, miR-10a-5p, miR-141-3p, miR-200c-3p, miR-182-5p, miR-183-5p, and miR-1206. OMIs significantly reduced migration for miR-183-5p, miR-203a, miR-296-5p, and miR-1206. Molecular pathway analysis revealed that the nine miRNAs regulate pathways associated with proliferation, invasion, and chemoresistance through PTEN, ZEB1, FOXO1, and SNAI2. High expression of miR-1206, miR-10a-5p, miR-141-3p, and miR-96-5p correlated with poor prognosis in OC patients according to the KM plotter database. These nine miRNAs could be used as targets for therapy and as markers of cisplatin response.

## 1. Introduction

Globally, an estimated 300,000 people are diagnosed with ovarian cancer each year, which causes approximately 180,000 deaths. Around 90% of ovarian tumors arise in epithelial cells of the ovary and/or in the fallopian tubes [1,2,3]. Histologically, epithelial ovarian cancer (EOC) is classified as serous, endometrioid, clear cell, or mucinous. Serous ovarian cancer represents approximately 70% of epithelial tumors. According to the proliferation rates, serous ovarian cancer can be further classified as low-grade serous ovarian cancer (LGOSC) or high-grade serous ovarian cancer (HGSOC). Estimates indicate that three out of four ovarian cancers are HGSOC. The first-line treatment for women with ovarian cancer includes surgery with pre- or postoperative platinum/taxane combined chemotherapy. Although most HGSOC cases are very sensitive to platinum chemotherapy, more than 70% of women become resistant to chemotherapy. Unfortunately, there is a lack of optimal therapeutic options for women experiencing recurrence. Hence, there is an urgent need for improved therapies targeting advanced and drug-resistant ovarian cancer.

MicroRNAs (miRNAs) are small noncoding RNA molecules that regulate gene expression at the posttranscriptional level by complementary binding to messenger RNAs (mRNAs) and subsequently degrading them or inhibiting their translation [4,5]. A single miRNA can simultaneously regulate multiple target genes [5]. The dysregulation of miRNAs promotes all steps of the carcinogenesis process, including cancer cell growth, proliferation, metastasis, and drug resistance [4]. Several miRNA profiling studies have identified aberrantly expressed miRNAs in ovarian cancer cell lines and ovarian tumor tissues [6,7]. However, comparisons of differentially expressed miRNAs between ovarian tumor samples and cisplatin-sensitive and cisplatin-resistant HGSOC cells have not been performed. In the present study, we assessed miRNA expression profiles in formalin-fixed, paraffin-embedded (FFPE) samples from patients with recurrent ovarian cancer and in cisplatin-sensitive and cisplatin-resistant OV-90 and OVCAR3 HGSOC cells. Nine miRNAs were commonly upregulated in both ovarian cancer samples and cell lines. Although most of the nine miRNAs have been studied individually, studies comparing at the same time the biological effects of targeting these miRNAs with oligonucleotide miRNA inhibitors (OMIs) have not been evaluated. OMIs significantly inhibited colony formation and reduced invasion in HGSOC cells. Interrogation of the Kaplan–Meier (KM) plotter patient databases revealed correlations between overall survival (OS) and miRNA expression for some of the nine miRNAs. Interrogation of available internet databases and literature reports revealed key genes regulated by various of the nine miRNAs. These findings strongly suggest that these identified miRNAs could serve as promising therapeutic targets for cisplatin-resistant HGSOC.

## 2. Results

### 2.1. MiRNA Profiles in Ovarian Cancer FFPE Samples and HGSOC Cells

The NanoString platform was used on RNA from FFPE samples and cisplatin-resistant ovarian cancer cells (OV-90CIS and OVCAR3CIS) and their sensitive counterparts (OV-90 and OVCAR3). Initially, 788 miRNAs were identified in ovarian cancer FFPE tissue samples vs. controls (Appendix A). Using a significance level of *p*-value < 0.05 resulted in a list of 76 differentially abundant miRNAs in the FFPE samples (Table 1). Similarly, 798 miRNAs were identified in cisplatin-resistant (OVCAR3CIS and OV-90CIS) cells compared with cisplatin-sensitive (OVCAR3 and OV-90) cells (Appendix A). A total of 40 deregulated miRNAs were identified after 0.05 *p*-value filtering (Appendix A), and after eliminating eleven miRNAs that exhibited opposite tendencies between the two cell lines, a total of 29 miRNAs were differentially expressed between cisplatin-sensitive and cisplatin-resistant cells (Table 2). We then selected the miRNAs commonly regulated in patient samples and in HGSOC cells. As shown in the Venn diagram in Figure 1, ten miRNAs were found to be common to both lists; nevertheless, miR-221-3p exhibited opposite effects on cell lines and FFPE patient tissues and was therefore excluded from subsequent studies. Curiously, all nine miRNAs were upregulated in patient samples compared with control samples and in cisplatin-resistant cells compared with cisplatin-sensitive cells. Table 3 includes the list of the nine miRNAs with their reported expression patterns and biological roles.

### 2.2. Effects of miRNA Inhibitors (OMIs) on Cell Proliferation and Invasion

The biological consequences of targeting most of these nine miRNAs have already been studied in ovarian cancer, and many of them have been proposed as diagnostic and prognostic markers and as targets for ovarian cancer therapy [10,32,33,34,35,36,37]. However, comparisons of the biological consequences of the knockdown of these nine miRNAs under the same experimental conditions have not been performed in HGSOC cells. To bridge this gap, we transiently transfected OVCAR3CIS cells with OMIs against each of the nine miRNAs and performed colony formation and invasion assays. We measured the effect of knocking down each miRNA on cell proliferation by clonogenic assays. Compared with Negative Control–OMI (NC-OMI), transient transfection of miRNA OMI (100 nM final concentration) significantly reduced the number of colonies for all miRNAs (Figure 2a). In particular, targeting miR-10a-5p, miR-96-5p, miR-141-3p, miR-203a, and miR-1206 reduced the number of colonies by 63% (*p* ≤ 0.0001), 69% (*p* ≤ 0.0001), 63% (*p* ≤ 0.0001), 66% (*p* ≤ 0.0001), and 61% (*p* ≤ 0.0001), respectively; targeting miR-182-5p and miR-200c-3p reduced the number of colonies by 51% (*p* ≤ 0.001); and targeting miR-296-5p reduced the number of colonies by 36% (*p* ≤ 0.01); all of these results were compared with those obtained with NC-OMI (Figure 2a). Next, we studied the effects of miRNA OMIs on reducing the invasion ability of OVCAR3CIS cells using transwell migration assays. The miRNA OMI inhibitors (50 nM final concentration) significantly reduced the invasion ability of four out of the nine miRNAs tested. Compared with NC-OMI, miR-183-5p, miR-203a, miR-1206, and miR-296-5p reduced the invasion ability of the cells by 79% (*p* ≤ 0.0001), 54% (*p* ≤ 0.01), 61% (*p* ≤ 0.01), and 58% (*p* ≤ 0.01), respectively (Figure 2b).

### 2.3. Bioinformatic Analysis of miRNA-Regulated Genes

By using the miRTargetLink 2.0 database, we identified genes that are regulated by each of the nine miRNAs (Appendix A). This tool allowed us to generate interactive networks for each of the nine miRNAs (Appendix A). Table 4 summarizes the genes regulated by four, three, and two (interactors) of the nine miRNAs according to miRTargetLink. For example, PTEN is potentially regulated by miR-200c-3p, miR-141-3p, miR-182-5p, and miR-10a-5p. By using Mienturnet (microRNA ENrichment TUrned NETwork), we generated an additional list of candidate genes for each of the nine miRNAs (Appendix A). Table 5 shows the genes regulated by five and four miRNAs as per Mienturnet. According to this tool, ZEB1 and IGF1R are potentially regulated by five of the nine miRNAs included in our study. We combined the information from the miRTargetLink 2.0 and miRTarBase databases (Table 4 and Table 5) to generate an interaction pathway of the genes regulated by the nine miRNAs (Figure 3a). Using the two databases PTEN, ZEB1, FOXO1, and SNAI2 are potentially regulated by four of the nine miRNAs. The genes regulated by both the Mienturnet and miRTargetLink 2.0 programs are listed in Appendix A.

### 2.4. Validation of the miRNA-Predicted Genes by Western Blots

To confirm whether targeting each of the miRNAs with OMIs increased the protein levels of the miRNA-predicted genes, we performed Western blot analysis. We focused these experiments on PTEN, ZEB1, FOXO1, and SNAI2, which are potentially regulated by four of the nine studied miRNAs. The Western blots shown in Figure 3b indicate that compared with those in the NC-OMI, the PTEN protein levels increased in the OVCAR3CIS cells transfected with each of the four miRNA-OMIs (miR-10a-5p, miR-141-3p, miR-182-5p, and miR-200c-3p), as predicted by the bioinformatic analysis. The protein levels of FOXO1 increased only with miR-183-5p OMIs. The protein levels of ZEB1 and SNAI2 did not increase with any of the miRNA-OMIs. Densitometric analysis of the band intensities confirmed these findings (Figure 3b).

### 2.5. Kaplan-Meier Survival Analysis

Kaplan-Meier (KM) patient survival analysis was performed using the Kaplan-Meier plotter database (accessed September 2023). For these studies, *p*-values < 0.05 were considered to indicate statistical significance. We found that the miRNA expression of miR-1206, miR-96-5p, miR-10a-5p, and miR-141-3p significantly correlated with the OS of ovarian cancer patients (Figure 4). That is, ovarian cancer patients with higher levels of these four miRNAs live less than ovarian cancer patients with lower levels of these miRNAs. There was no statistically significant difference in overall survival (OS) or the expression levels of miR-182-5p, miR-296-5p, miR-200c-3p, miR-183-5p, or miR-203a (Figure 4).

## 3. Discussion

In this study, we conducted a comprehensive analysis of miRNA expression profiles in formalin-fixed paraffin-embedded (FFPE) ovarian human samples and cell lines derived from High-Grade Serous Ovarian Carcinoma (HGSOC). We identified significantly increased levels of nine miRNAs associated with cisplatin resistance including miR-10a-5p, miR-96-5p, miR-141-3p, miR-182-5p, miR-183-5p, miR-200c-3p, miR-203a, miR-296-5p, and miR-1206. Although we used a small number of patient samples, the combination of patient data with the use of cisplatin-sensitive and cisplatin-resistant HGOSC cells increased the reliability of our results, as most of these miRNAs have been previously reported as key players in ovarian cancer initiation, progression, and drug resistance. Similarly, although the roles of these miRNAs have been reported, comparisons of the biological consequences of targeting these nine miRNAs under the same experimental conditions have not been previously reported.

Limited information in the literature is available regarding the role of miR-1206 in ovarian cancer or other cancer types. Gutierrez-Camino et al. showed that miR-1206 is a potential biomarker for predicting methotrexate toxicity during chemotherapy treatment in pediatric acute lymphoblastic leukemia patients [30]. qRT-PCR of 36 pairs of tumor tissues and adjacent tissues collected from lung cancer patients revealed increased miR-1206 expression in the cancerous tissues [31]. Additionally, in the A549 and PC9 lung cancer cell lines, miR-1206 expression was greater than that in their noncancerous counterparts [31]. Inhibition of miR-1206 suppressed cell proliferation, blood vessel formation, migration, and invasion but increased apoptosis in A549 and PC9 lung cancer cell lines [31,38]. We observed a prominent reduction in the proliferation and invasion of OVCAR3CIS cells upon miR-1206 knockdown, and its reduced expression correlates well with ovarian cancer patient outcomes. Our bioinformatic analysis identified more than 20 potential miR-1206 targets, including E2F3 (a transcription factor that regulates cell proliferation) [39] and TP53INP1 (a tumor suppressor gene regulating autophagy) [40]. According to our results, miR-1206 represents an important target for ovarian cancer therapy; therefore, further research is required to identify the miR-1206 targets responsible for the observed biological effects.

MiR-96-5p is an oncomiR that is overexpressed in various cancer types, including cervical, head and neck, lung, breast, prostate, thyroid, colorectal, and ovarian cancer [11,41]. Using luciferase reporter assays, Ling Li et al. reported that SNAI2 and ZEB1 are miR-96 target genes in HCT116-p21-/- colon carcinoma-derived cells [42]. ZEB1 is a transcription factor that promotes tumor invasion and metastasis through the regulation of genes that promote EMT [42]. A study by Cui et al. showed that downregulation of ZEB1 can effectively decrease the proliferative and invasive capacities of SKOV3/DPP cisplatin-resistant ovarian cancer cells upon cisplatin treatment and that in vivo downregulation of this miRNA effectively decreases the tumor volume and weight of nude mice [43]. More than 15 additional miR-96-5p target genes, including CAVEOLAE1, FOXO1, and FOXO3A, have been reported [41,42]. Our bioinformatic analysis predicted that miR-96-5p regulates ZEB1, FOXO1, and SNAI2, among others. However, in our Western blots, we did not observe increased expression of any of these three potential miR-96-5p target genes. Liu et al. used miRNA mimics to increase miR-96-5p in SKOV3 and CAOV3 ovarian cancer cell lines and observed increased levels of cell proliferation, colony formation, and migration [11]. In our study, knockdown of miR-96-5p resulted in significant reductions in cell proliferation but not in the invasive ability of these cells. However, Kaplan-Meier curves showed that high miR-96 expression levels were correlated with a decrease in OS of ovarian cancer patients. Additional studies are required to identify the target genes of miR-96-5p in ovarian cancer.

Liu et al. used qRT-PCR and reported that miR-10a-5p is downregulated in human ovarian cancer tissues and ovarian cancer cells compared with their normal counterparts. HOXA1 was significantly decreased in ovarian cancer cells after transfection with miR-10a-5p mimics, and its overexpression abrogated the effect of miR-10a-5p, leading to a reduction in cell viability and migration [8]. In contrast, Wang et al. reported that miR-10a-5p is overexpressed in the plasma of ovarian cancer patients but is expressed at lower levels in tumor samples than in normal controls [44]. We observed increased expression in ovarian cancer tissue samples and cisplatin-resistant HGSOC cell lines compared with controls. Additionally, our Kaplan-Meier curves showed that miR-10a high expression levels were correlated with poor ovarian cancer patient outcomes. Notably, Benson et al. demonstrated substantial variability in the plasma concentrations of miR-10a-5p among recurrent platinum-resistant ovarian cancer patients [45]. This variation was found to be linked to the response to decitabine followed by carboplatin chemotherapy, suggesting that miR-10a-5p could serve as a therapeutic response target [45]. Additionally, studies have highlighted that miR-10a inhibition is associated with sensitivity to genotoxic chemotherapeutics, such as gemcitabine, temozolomide, and cisplatin [46]. For instance, in lung carcinoma, Huang et al. showed that miR-10a enhances cisplatin resistance by suppressing the PI3K/Akt pathway [47]. Moreover, Sun et al. demonstrated that miR-10a downregulation decreased cisplatin resistance in A549 cells via the TGF-β/smad2/STAT3/STAT5 signaling pathway [48]. In our study, we observed a strong reduction in cell proliferation upon miR-10a-5p knockdown. However, the same treatment did not significantly reduce the invasion of OVCAR3CIS cells, which indicates that miR-10a-5p regulates genes involved in cell growth and proliferation. Our bioinformatic analysis predicted that miR-10a-5p regulates PTEN, BDNF, and XIAP. Moreover, our Western blot analysis confirmed that reduced expression of miR-10a-5p increased the level of PTEN. Yu et al. found that miR-10a is elevated in non-small cell lung cancer (NSCLC) and that its expression correlated with the progression of NSCLC by targeting PTEN [49]. As this miRNA is a promising therapeutic target in cisplatin-resistant ovarian cancer, further research is needed to elucidate additional miR-10a-5p targets.

Contrasting reports have been published regarding the role of miR-141-3p as either an oncomiR or a tumor suppressor miRNA [50]. Elevated levels of miR-141-3p have been linked to poorer survival and the promotion of cancer in diverse cancer types, including breast, colorectal, bladder, and ovarian cancer. We also observed a significant correlation between high miR-141-3p levels and decreased OS in ovarian cancer patients. Additionally, in prostate and esophageal cancers, miR-141 overexpression is associated with an increased risk of recurrence, distant metastasis, and worse survival outcomes [51]. However, it can act as a tumor suppressor in gastric and colorectal cancers, inhibiting proliferation and migration [52]. In ovarian cancer, Jaarsveld et al. compared the miRNA expression profiles of cisplatin-sensitive and cisplatin-resistant ovarian cancer cell line pairs. Their findings revealed 27 miRNAs with differential expression (≥2-fold), and notably, five of these, including the family members miR-141 and miR-200c, exhibited a correlation with cisplatin sensitivity. Overexpression of miR-141 in the sensitive cell line A2780 increased cisplatin resistance by targeting the NF-κB pathway [53]. Additionally, Mateescu et al. showed that miR-141, through direct targeting of p38α, promotes ovarian tumorigenesis in mouse models by controlling the oxidative stress response [54]. Zhong et al. performed luciferase reporter assays in papillary thyroid cancer and identified PTEN as a miR-141 target gene [55]. Our bioinformatic studies also identified PTEN as an miR-141-3p target, and Western blot analysis revealed increased PTEN protein levels following miR-141-3p knockdown. Additionally, we found that knockdown of miR-141-3p in OVCAR3CIS cells strongly reduced cell proliferation but not cell invasion. Nevertheless, further in vivo experiments are required before this miRNA can be proposed as a plausible target for ovarian cancer treatment.

MiR-183-5p dysregulation has been associated with tumors, including lung, ovarian, breast, bladder, and prostate cancer, and has been implicated in tumor invasion and metastasis [56]. Bioinformatic analysis by Sarver et al. confirmed that miR-183 is upregulated in synovial sarcoma, rhabdomyosarcoma, and colon cancer tumors compared with normal tissues [57]. This study also showed that miR-183 affects both EGR1 mRNA transcript and EGR1 protein levels via translational regulation in these tumors. Mohammaddoust et al. performed gene expression studies using NCBI-GEO databases and reported that miR-183 is increased in human breast tumors compared with normal tissues [58]. In addition, dual luciferase and Western blot assays revealed that PTEN is a direct target of miR-183 [58]. These findings suggest that miR-183 acts as an oncogene, promoting cell viability (as assessed by the MTT assay), migration (as assessed by wound healing and transwell migration assays), and cell cycle progression (as assessed by flow cytometry). In the context of ovarian cancer, Zhou et al. reported significant upregulation of miR-183 in tumor samples and SKOV-3 and OVCAR3 cell lines compared with normal tissues and normal human ovarian surface epithelial (HOSE) cells, respectively [19]. Functional assays showed that inhibiting miR-183 led to a significant reduction in cell proliferation and invasion while inducing apoptosis, particularly through the TGF-β/Smad4 signaling pathway [19]. Moreover, Chen et al. also showed that miR-183 was aberrantly overexpressed in epithelial ovarian cancer tumors compared with their normal counterparts and that its expression was greater in patients at advanced clinical stages [20]. They also showed that downregulation of miR-183 led to reduced cell viability and proliferation in SKOV3 and ES-2 cells [20]. Our bioinformatic analysis predicted that miR-183-5p regulates ZEB1, FOXO1, and SNAI2, among other targets. Interestingly, Li et al. showed that in colorectal cancer cell lines, the miR-183-96-182 cluster represses ZEB1, a master regulator of epithelial to mesenchymal transition (EMT) [42]. Our Western blot results agreed with these reports, as we observed increased expression of FOXO1 and SNAI2 upon miR-183-5p knockdown. FOXO1 plays a critical role in diverse cancer-related processes, including cellular differentiation, apoptosis, cell cycle arrest, and the response to DNA damage [59], and is associated with poor prognosis in epithelial ovarian cancer patients [60,61]. On the other hand, SNAI2 is a strong E-cadherin repressor with a central role in EMT [62]. Studies by Jin et al. showed that SNAI2 knockdown suppressed cell migration and invasion and induced apoptosis by inducing ferroptosis in ovarian cancer cells [63]. Interestingly, knockdown of miR-183-5p resulted in the greatest reduction in the invasion ability and colony formation ability of OVCAR3CIS cells, which suggested that miR-183-5p is a prominent target against cisplatin-resistant ovarian cancer.

Xiaohong et al. performed a study using 30 samples of ovarian cancer tissue and adjacent noncancerous tissues and isolated total RNA (including miRNAs) and found by qPCR that the relative expression of miR-203 in ovarian cancer tissues was significantly greater than that in adjacent noncancerous tissues [24]. Inhibition of miR-203 in Ovca429 and Ovca433 cells resulted in reduced migration, impaired glucose consumption and lactate production, suggesting a role for this miRNA in glycolysis [24]. Zhang et al. used a miR-203a-3p mimic and observed reduced PTEN expression, and using luciferase reporter assays, they demonstrated that PTEN is an miR-203a-3p regulated gene in BRL-3A hepatocytic cells [64]. This evidence supports our findings, as we observed that miR-203a inhibition significantly reduced HGSOC cell growth and invasion. However, our bioinformatic studies did not identify PTEN as a potential target gene of miR-203a. More studies are required to identify the target genes of miR-203a in cisplatin-resistant HGSOC cells. Candidate genes included SNAI2 and E2F3, which were identified by the two databases and pathway analysis tools used in our bioinformatic studies.

Increased levels of miR-200c-3p have been observed in various cancer types, including breast, ovarian, prostate, endometrial, lung, colon, and pancreatic cancer [65]. In HGSOC, miR-200c plays a role in the metastasis and invasion of ovarian carcinoma by regulating EMT processes [66]. It has also been proposed as a biomarker for both HGSOC and non-HGSOC due to its presence in serum exosomes [66]. In our study, we also detected high miR-200c-3p levels in HGSOC tumors and cisplatin-resistant HGSOC cells. Although we observed a strong reduction in cell proliferation upon miR-200c-3p knockdown, the same treatment did not significantly reduce the invasive ability of the OVCAR3CIS cells. Several miR-200c target genes, including VEGF, ZEB1, ZEB2, and CDH1, have been identified [66]. Our bioinformatic analysis identified PTEN, FOXO1, and ZEB1 as three potential targets of miR-200c. However, according to Western blot analysis, only PTEN protein levels were increased following miR-200c-3p knockdown in OVCAR3CIS cells.

Dysregulation of miR-182-5p has been observed in many types of cancer, including gastric, colorectal, cervical, osteosarcoma, breast, glioma, and pancreatic cancer [67]. Contrasting results have also been reported regarding the role of this miRNA, as it can act both as a tumor suppressor and as an oncomiR [17,67]. MiR-182-5p has been associated with the modulation of various signaling pathways, such as the JAK/STAT3, Wnt/β-catenin, TGF-β, and P13K/AKT pathways [67]. Liu et al. showed that miR-182 was significantly overexpressed in ovarian patient tissue samples compared with normal tissue samples [17]. Overexpression of miR-182 resulted in accelerated tumor transformation and metastasis and increased invasion in vitro [17]. RT-PCR and Western blot studies have shown that HMGA2 and MTSS1 are regulated by miR-182 [17]. Moreover, Zhang et al. showed that miR-182-5p promotes cisplatin resistance in A2780 and SKOV3 epithelial ovarian cancer cells by targeting GRB2, as confirmed by luciferase reporter assays [68]. Our bioinformatics analysis identified PTEN, FOXO1, and SNAI2 as potential targets of miR-182-5p. Our Western blot assays revealed that miR-182-5p inhibition increased the protein levels of PTEN but not FOXO1 or SNAI2.

Evidence indicates that miR-296-5p can act either as an oncomiRNA (oncomiR) or a tumor suppressor [27]. Many studies have emphasized the function of microRNA-296 (miR-296) that inhibits tumor formation. To some extent, the role of miR-296 in esophageal squamous cell carcinoma (ESCC) remains misleading. Therefore, the current research was designed to investigate the regulatory mechanisms of miR-296 and signal transducer and activator of transcription 3 (STAT3) in ESCC. PATIENTS AND METHODS: The mRNA expression of miR-296-5p and STAT3 in ESCC tissues or cell lines was measured via quantitative Real Time-Polymerase Chain Reaction (qRTPCR). The protein level of STAT3 was measured by Western blotting assay. The luciferase reporter assay was used to verify the binding sites between miR-296-5p and STAT3. The transwell assay was employed to identify cell migration and invasion. RESULTS: Down-regulation of miR-296-5p was detected in ESCC tissues and cell lines (*p* < 0.01). Additionally, miR-296-5p was found to target STAT3 directly. Functionally, up-regulation of miR-296-5p or down-regulation of STAT3 significantly inhibited cell migration and invasion in ESCC. CONCLUSIONS: MiR-296-5p inhibited cell invasion and migration in ESCC by downregulating STAT3. The overexpression of miR-296-5p by targeting STAT3 suppressed tumorigenesis of ESCC cells [27]. High miR-296 expression levels have been associated with cancer promotion in colorectal cancer [38] and aggressive cancers such as glioblastoma (GBM) [69]. In GBM, increased levels of miR-296-5p enhanced the invasive properties of cells through the downregulation of caspase-8 and nerve growth factor receptor (NGFR), which were identified as miR-296-5p targets through bioinformatic analysis [69]. In contrast, in epithelial ovarian cancer (EOC), increased expression of miR-296 was associated with decreased levels of EMT-related proteins [26]. Yan et al. identified miR-296 as an upstream regulator of the calcium-binding protein S100A4. This protein promotes cancer progression by modulating the expression of genes such as Snail and MMP9, two key proteins regulating EMT [26]. In SKOV-3 and HO-8910PM cells, downregulation of miR-296 decreased the mRNA levels of S100A4 and the levels of EMT-related proteins. This information indicates that in epithelial ovarian cancer, miR-296 has an antitumorigenic effect [26]. According to our results in HGSOC, miR-296-5p acts as an oncomiR, as its knockdown reduced the proliferation and invasion of OVCAR3CIS cells. Therefore, further studies are needed to identify the miR-296 target genes that account for the observed biological effects. Interestingly, our bioinformatic analysis identified Notch and VEGFA as potential direct targets of miR-296-5p.

To summarize, we identified a signature of nine miRNAs in the tissues of recurrent cisplatin-treated HGSOC tumors and cisplatin-resistant HGSOC cell lines compared to their noncancerous and cisplatin-sensitive counterparts, respectively. Future studies include the use of ovarian cancer mouse models to test the efficacy of the OMIs against these nine miRNAs. Other miRNAs that were highly abundant in tumor tissues included miR-200a-3p and miR-135b-5p, which have been previously associated with OC promotion, and miR-98-3p, which could be a potential target in OC [61,62]. Therefore, additional studies with a greater number of human samples could identify additional miRNAs deregulated in cisplatin resistant HGSOC.

## 4. Materials and Methods

### 4.1. Cell Lines and Cell Culture Maintenance

The human high-grade serous ovarian cancer cell lines OV-90 and OVCAR3 were purchased from the American Type Culture Collection (ATCC, Manassas, VA, USA). OV-90CIS and OVCAR3CIS cells were generated by exposing parental cell lines to increasing doses of cisplatin as previously described [70]. OV-90 and OV-90CIS cells were maintained in M199 (Gibco, Thermo Fisher Scientific, Grand Island, NY, USA)/MCDB-105 (Sigma-Aldrich, St. Louis, MO, USA) media. OVCAR3 and OVCAR3CIS cells were maintained in RPMI-1640 (Thermo Scientific, Grand Island, NY, USA) medium supplemented with 0.01 mg/mL insulin (Sigma-Aldrich). In all cases, the medium was supplemented with 10% fetal bovine serum (FBS; HyClone, GE Healthcare Life Sciences, Logan, UT, USA) and 0.1% antibiotic/antimycotic solution (HyClone). The cells were incubated at 37 °C in 5% CO_2_ with 95% air. All cell lines were screened for mycoplasma using the LookOut^®^ Mycoplasma PCR detection kit (Sigma). In vitro experiments were performed at 70–85% cell confluence.

### 4.2. Description of Patient Samples and RNA Isolation from Paraffin Samples and Cell Lines

Formalin-fixed, paraffin-embedded (FFPE) tissue blocks from newly diagnosed (de novo) ovarian cancer patients were obtained from the Pathology Department of the University of Puerto Rico Medical Sciences Campus. The research protocol was approved by the University of Puerto Rico Medical Sciences Campus Institutional Research Board. In this study, we included five serous ovarian cancer samples from patients who experienced recurrence within a year following their last cisplatin therapy (aged 43–77 years, median age: 55 years). To serve as controls, six FFPE ovarian samples from normal subjects (aged 39–65 years; median age, 49 years) were selected based on specific criteria, namely, nonneoplastic and noninfectious disease. The samples were kindly provided by the Department of Puerto Rico Medical Sciences Campus. A qualified pathologist reviewed representative hematoxylin and eosin (H&E)-stained slides from each tissue block to accurately delineate the tumor areas. For each FFPE tissue block, a 3 mm punch biopsy sample was obtained from a tumor area or from ovarian tissue (controls). Each sample was identified with a numeric code (no personal identifiers were employed). The samples were placed in a microcentrifuge tube for miRNA extraction and processed using the RecoverAll Total Nucleic Acid Isolation Kit (Fisher, Austin, TX, USA) according to the manufacturer’s instructions. Briefly, the samples were subjected to deparaffinization with 100% xylene followed by washing with 100% ethanol, after which they were dried in a centrifugal vacuum concentrator at 40 °C. A mixture of 300 μL of digestion buffer and 6 μL of protease was added to each sample, which was then incubated at 50 °C for 3 h, heated to 85 °C for 15 min, and left at −20 °C overnight. Subsequently, 200 μL of isolation additive was added to the samples, which were then vortexed and mixed with 550 μL of ethanol. The final mixture was passed through a filter cartridge and centrifuged at 10,000× *g* for 30 s. The samples were washed once with wash solution 1 and once with wash solution 2/3. Each sample was then subjected to DNase digestion for 1 h. The samples were subjected to a second consecutive washing step with wash solution 1 and wash solution 2/3. The RNA was eluted with 60 μL of dH_2_O, followed by centrifugation at maximum speed for 1 min. The RNA concentration and quality were determined using a Thermo Scientific NanoDrop spectrophotometer.

Total RNA from ovarian cancer cell lines was isolated using the mirVana miRNA Isolation Kit (Thermo Fisher Scientific) following the manufacturer’s instructions. Briefly, pellets of cells were lysed by adding lysis/binding solution and vortexing. Subsequently, the miRNA homogenate additive (1:10 lysate volume) was added to the lysate, mixed by inverting the tube, and then left on ice for 10 min. Following incubation, Acid-Phenol:chloroform was added, and the samples were mixed and centrifuged at room temperature. The aqueous phase was removed, the sample was transferred to a fresh tube, and 100% ethanol was added. The lysate/ethanol mixture was added to a filter cartridge, centrifuged, and washed. Total RNA was eluted with boiling water. The RNA concentration and quality were determined using a Thermo Scientific NanoDrop spectrophotometer.

### 4.3. MiRNA Expression Profiles Using NanoString Technology

Total RNA was diluted to 33 ng/µL, and miRNA expression profiles were assessed using the nCounter Analysis System from NanoString Technologies (https://www.nanostring.com/products/mirna-assays/mirna-panels, accessed on 23 March 2023), utilizing 100 ng of total RNA. The NanoString nCounter Human v2 panel contains 798 unique miRNA barcodes of well-characterized human miRNAs, each equipped with a capture probe and a reporter probe, alongside a sequence that complements the target sequence. The capture probe is enriched with a biotin molecule, facilitating sample immobilization, while the reporter probe carries a distinctive fluorescent barcode for precise detection. The reporter probes and capture probes are then hybridized to the sample. After the capture and reporter probes are bound to the mRNA, a stable tripartite structure is formed and linked to a streptavidin-coated cartridge (via the biotin capture probe), and alignment is achieved electrophoretically. The barcodes are then counted and tabulated. The digital analyzer uses an epi-fluorescence microscope with an oil immersion lens and a charge-coupled device (CCD) camera to collect image data, which are subsequently converted into a digital signal. The brightness of the code does not carry any information; the probe is simply considered to be present or not present. The data output is simply the gene name, accession number, and number of times that the transcript was counted in that sample.

### 4.4. Transient Transfection of Oligonucleotide miRNA Inhibitors (OMIs)

One day before transfection, the cells (3.0 × 10^4^ cells/mL) were plated in 10-cm Petri dishes. The next day, OMIs (MirVana RNA inhibitors, Thermo Fisher Scientific) were mixed with Lipofectamine RNAiMAX Transfection reagent (Thermo Fisher Scientific) at a 1:2 volume ratio (OMIs:Lipofectamine) and incubated for 15–20 min in serum and antibiotic-free Opti-MEM at room temperature. We used a negative control (NC) (miRVana miRNA Inhibitor, Negative Control #1, Thermo Fisher Scientific) for comparison. The cell culture media of the cells was replaced with Opti-MEM, and the transfection mixture was added dropwise. The transfected cells were incubated overnight, after which the cell pellets were collected for subsequent experiments.

### 4.5. Clonogenic and Migration Assays

Cell proliferation was assessed with clonogenic assays. Cells (3.0 × 10^4^ cells/mL) were seeded into 6-well plates, and 24 h later, the cells were transfected with OMIs (100 nM final concentration) as described above. The next day, the transfected cells (1000) were seeded in 10 cm Petri dishes. After seven days, the colonies that had formed were stained with 0.5% crystal violet in methanol. Colonies of at least 50 cells were quantified under a light microscope (CKX41; Olympus) at 10× magnification in five random fields. The percentages of clonogenicity were calculated relative to the control. To assess cell invasion, cells (2 × 10^4^ cells/mL) were seeded in 10 cm Petri dishes. Twenty-four hours later, the cells were transfected with OMIs (50 nM final concentration). The next day, 70,000 cells were seeded into Matrigel-coated transwells. Forty-eight hours later, the cells were fixed and stained using the Fisher HealthCare™ PROTOCOL™ Hema 3™ Manual Staining System. The invading cells were counted at 20× on an Olympus 1 × 71 microscope equipped with a digital camera (Olympus DP26). The percentages of invaded cells were calculated, taking the untransfected cell values as 100% of the cell invasion.

### 4.6. Bioinformatic Analysis for miRNA Target Prediction and Pathway Construction

We used miRTargetLink 2.0 and the Mienturnet interactive miRNA target gene and target pathway networks. miRTargetLink 2.0 was used for the miRTarBase, mirDIP, miRDB, and miRATBase published miRNA repositories. miRTargetLink 2.0 offers information on miRNA-mRNA interactions and produces an interactive network that shows which miRNAs regulate a particular gene [71]. Mienturnet (microRNA ENrichment TUrned NETwork) uses computationally predicted and experimentally validated miRNAs, filters those miRNAs, and then performs a statistical and network-based analysis to assess the significance of miRNA-target interactions [72]. We combined the information from the miRTargetLink 2.0 and Mienturnet databases to determine the interaction pathways of the common genes regulated by each of the nine miRNAs.

### 4.7. Western Blot Analysis

Following transient transfection of OMIs, the cells were detached with trypsin (0.25%) at 37 °C, washed with phosphate-buffered saline (PBS), harvested, and stored at −80 °C until processing. Cells were lysed with ice-cold lysis buffer and incubated on ice for 30 min. Whole-cell lysates were centrifuged, supernatants were collected, and protein concentrations were determined using Bio-Rad Protein Reagents (Bio-Rad). In all cases, protein lysates (30–50 µg) were separated by SDS-PAGE (12% acrylamide), blotted onto nitrocellulose membranes, and probed with appropriate dilutions of primary antibodies against PTEN (Cell Signaling, Danvers, MA; 9569P), ZEB1 (Novus, Centennial, CO; 23484SS), FOXO1 (Cell Signaling 2880S), and SNAI2 (Cell Signaling 9585S). β-Actin was used as a loading control (Sigma Aldrich A5441). The membranes were rinsed and incubated with an anti-rabbit horseradish peroxidase-conjugated secondary antibody (Cell Signaling). Bound antibodies were detected using enhanced chemiluminescence (GE Healthcare, Chicago, IL, USA) followed by autoradiography in a ChemiDoc Imaging System (Bio-Rad).

### 4.8. Kaplan–Meyer (KM) Survival Analysis

Kaplan–Meyer (KM) plots analysis was performed using the publicly available KM plotter database (www.kmplot.com, accessed on 20 August 2023). We used the “Start miRpower for pan-cancer” window [73]. In this window, we selected 486 ovarian cancer samples. Here, for each miRNA symbol (i.e., hsa-miR200c), a probe ID was selected, and the ovarian cancer patients were categorized into high- or low-expression groups based on the median RNA expression values of the dataset. KM survival plots for overall survival (OS) were generated with their respective hazard ratios (HRs), confidence intervals (CIs), and *p*-values (log-rank). *p*-values < 0.05 were considered to indicate statistical significance.

### 4.9. Statistical Analysis

All experiments were analyzed using GraphPad Prism 8 (GraphPad Software, La Jolla, CA, USA). The NanoString experiments with ovarian cancer cells were performed in triplicate. Statistical differences were determined using a 2-tailed, unpaired Student’s t-test, and one-way and two-way ANOVA were performed as per the requirements of the analysis. * *p* ≤ 0.05, ** *p* ≤ 0.01, *** *p* ≤ 0.001, **** *p* ≤ 0.0001. A *p*-value less than 0.05 was considered to indicate statistical significance.

## 5. Conclusions

Our study identified a signature of nine miRNAs, all of which were increased in cisplatin-resistant ovarian cancer. MiR-183-5p, miR-203a-3p, miR-296-5p, and miR-1206 offer promising avenues for targeted therapies in HGSOC treatment. The nine miRNAs included in this study could be investigated for their prognostic value and response to therapy in HGSOC patients.

## Figures and Tables

**Figure 1 ijms-25-03793-f001:**
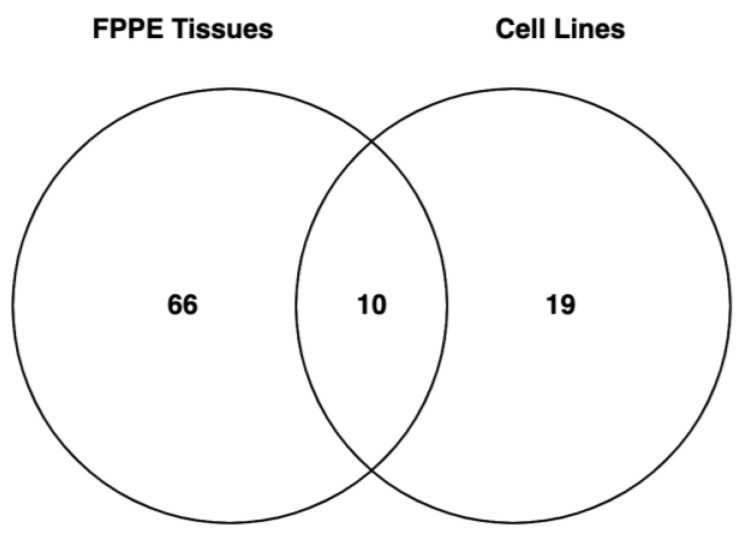
Venn diagram showing the unique and common microRNAs in FFPE ovarian cancer patient samples and HGSOC cell lines. Sixty-six unique microRNAs were identified in FPPE tissue samples, 19 in HGSOC cell lines, and 10 in both groups. After excluding miR-221-3p, the remaining nine genes were selected for biological studies.

**Figure 2 ijms-25-03793-f002:**
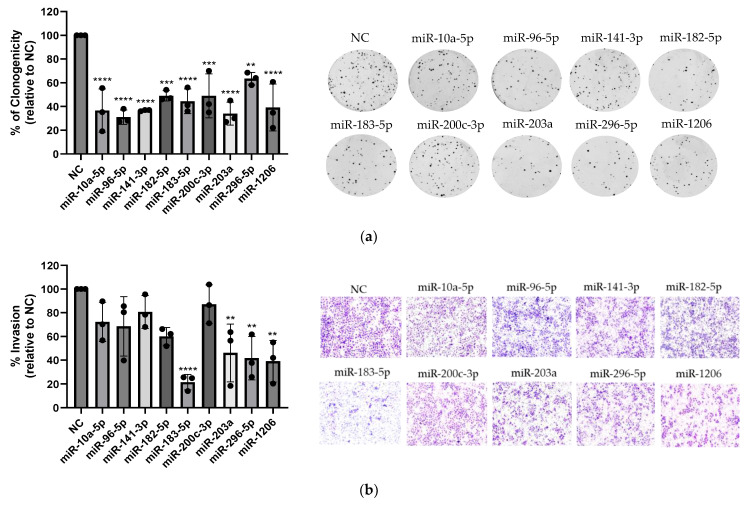
Effect of targeting miRNAs with OMI on cell proliferation and invasion. Each miRNA was knocked down with a specific OMI. (**a**) OVCAR3CIS cells were transiently transfected with 100 nM OMI to perform clonogenic assays as described in the Section 4. The % of clonogenicity was calculated relative to that of NC-OMI, which was taken as 100%. The experiments were repeated three times. Bars: mean ± SD (**** *p* < 0.0001, *** *p* < 0.0001, ** *p* < 0.01). Representative colonies following the transfection of OMIs are depicted on the right of the clonogenicity graph. Images were taken with a ChemiDoc Imaging System (Bio-Rad, Hercules, CA, USA). (**b**) OVCAR3CIS cells were transfected with 50 nM OMI to perform invasion assays. The % of invasion was calculated relative to NC-OMI, which was taken as 100%. Purple pores represent the Matrigel, while full purple circles represent cells. Bars: mean ± SD of triplicate experiments (**** *p* < 0.0001, ** *p* < 0.01). Representative images shown on the right of the invasion graph were taken at 20× magnification with an Olympus (Olympus, Center Valley, PA, USA) 1 × 71 microscope equipped with a digital camera (Olympus DP26).

**Figure 3 ijms-25-03793-f003:**
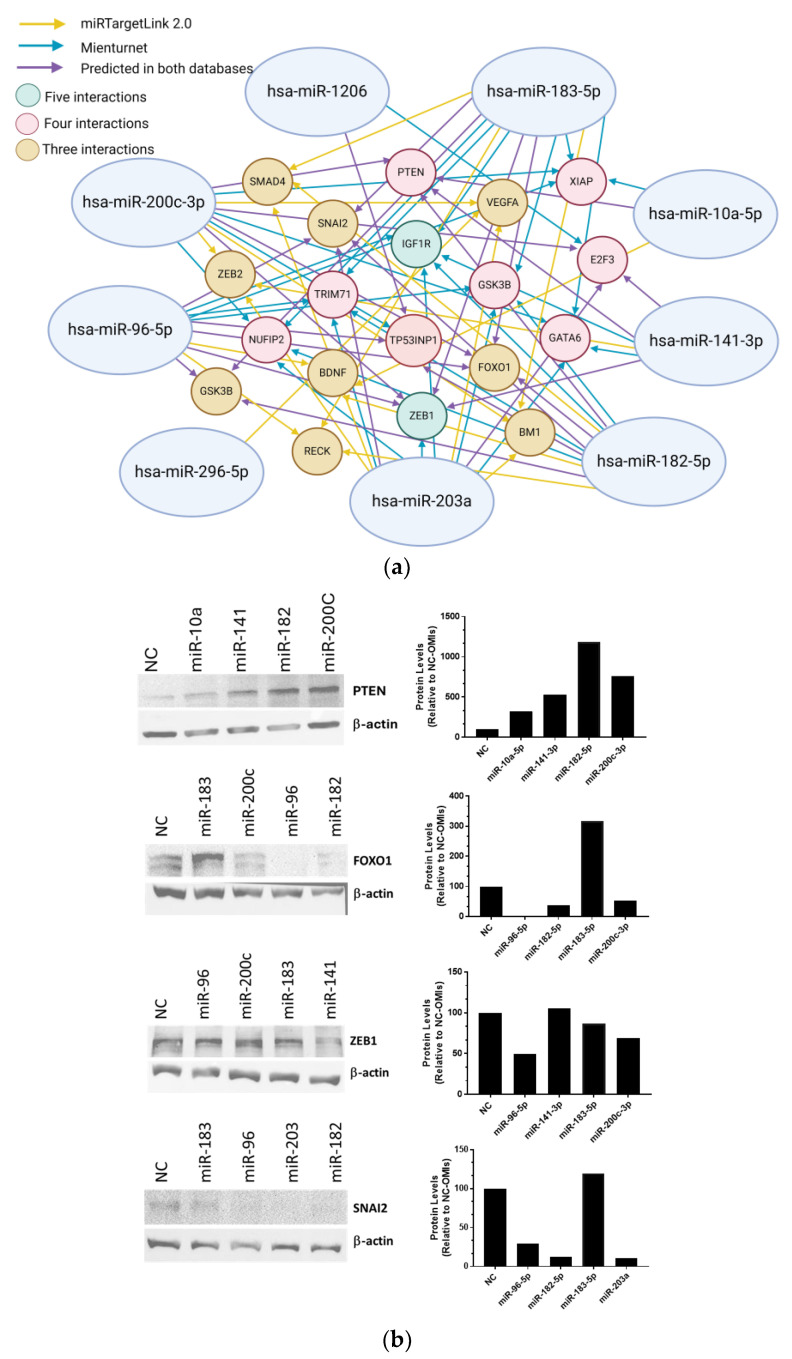
Interactions among the nine microRNAs and their potential target genes and validation of the miRNA-predicted genes at the protein level. (**a**) Interaction map of predicted miRNAs and target genes using miRTargetLink 2.0 and miRTarBase. Direct potential interactions are represented by colored solid lines: (yellow) interactions predicted by miRTargetLink 2.0, (blue) interactions predicted by miRTarBase, and (purple) interactions predicted commonly by both databases. The colored circles represent the number of predicted interactions of an miRNA with its target gene: (green) five interactions, (red) four interactions, and (gold) three interactions. (**b**) Western blot analysis with 30–50 μg of protein extracts following the transfection of OMIs in OVCAR3CIS cells. Densitometric analysis of the band intensities were calculated relative to β-actin and expressed relative to NC-OMIs.

**Figure 4 ijms-25-03793-f004:**
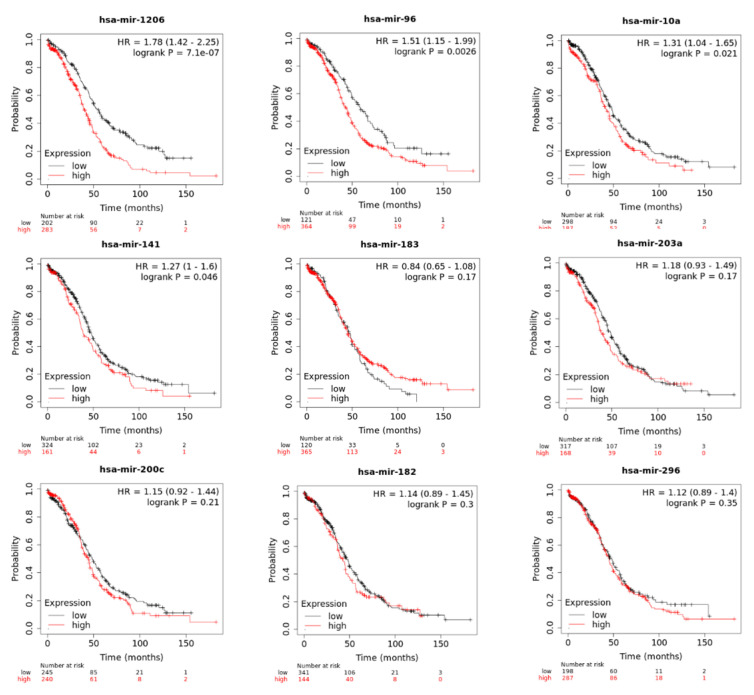
Kaplan–Meier survival analysis. Kaplan–Meier (KM) patient survival analysis was performed using the Kaplan-Meier plotter database in the “Start miRpower for pancancer” window. In this window we selected only 486 ovarian cancer samples. *p*-values < 0.05 were considered to indicate statistical significance.

**Table 1 ijms-25-03793-t001:** Differentially abundant miRNAs between ovarian cancer patient tissue samples and controls.

miRNA	Fold Change	*p*-Value
hsa-miR-200c-3p	33.0	0.00252
**hsa-miR-141-3p**	98.8	0.00436
hsa-miR-98-3p	2.39	0.00514
hsa-miR-200b-3p	37.6	0.00814
hsa-miR-135b-5p	170	0.00937
hsa-miR-320b	2.29	0.00946
hsa-miR-370-5p	2.00	0.01036
hsa-miR-106b-5p	12.4	0.01097
hsa-miR-3918	2.00	0.01170
hsa-miR-589-5p	2.83	0.01292
hsa-miR-512-3p	2.33	0.01355
hsa-miR-3928-3p	1.63	0.01417
hsa-miR-183-5p	6.19	0.01444
hsa-miR-93-5p	7.46	0.01588
hsa-miR-515-5p	1.83	0.01588
hsa-miR-577	1.81	0.01878
hsa-miR-625-5p	2.52	0.01906
hsa-miR-92b-3p	4.26	0.02038
hsa-miR-200a-3p	41.0	0.02094
**hsa-miR-96-5p**	6.08	0.02227
hsa-miR-182-5p	6.10	0.02455
hsa-miR-106a-5p-hsa-miR-17-5p	9.15	0.02534
hsa-miR-561-5p	2.20	0.02576
hsa-miR-619-3p	1.97	0.02578
hsa-miR-449a	4.89	0.02582
hsa-miR-663a	5.83	0.02603
hsa-miR-92a-3p	6.85	0.02633
hsa-miR-3180-5p	1.82	0.02675
hsa-miR-548d-3p	1.76	0.02793
**hsa-miR-1206**	3.07	0.02866
hsa-miR-484	3.10	0.02937
hsa-miR-651-5p	1.88	0.02942
hsa-miR-429	19.9	0.03255
hsa-miR-596	4.70	0.03294
hsa-miR-146b-5p	7.93	0.03301
hsa-miR-18a-5p	5.74	0.03313
hsa-miR-874-5p	2.61	0.03319
hsa-miR-610	2.01	0.03324
hsa-miR-203a-3p	10.0	0.03339
hsa-miR-20a-5p-hsa-miR-20b-5p	7.55	0.03394
hsa-miR-107	5.04	0.03397
hsa-miR-2117	1.98	0.03617
hsa-miR-371a-5p	1.91	0.03699
hsa-miR-323b-3p	1.76	0.03715
hsa-miR-1287-3p	1.91	0.03891
hsa-miR-221-3p	7.35	0.03926
hsa-miR-1244	1.48	0.03953
hsa-miR-548j-3p	1.67	0.04000
hsa-miR-30a-5p	4.02	0.04031
hsa-miR-3136-5p	1.75	0.04063
hsa-miR-25-3p	5.52	0.04103
hsa-miR-378i	4.71	0.04129
hsa-miR-205-5p	59.1	0.04227
hsa-miR-421	2.14	0.04259
hsa-miR-885-5p	5.30	0.04286
hsa-miR-563	1.55	0.04344
hsa-miR-181b-5p-hsa-miR-181d-5p	2.61	0.04368
hsa-miR-378 g	3.56	0.04439
hsa-miR-935	1.73	0.04492
hsa-miR-320c	1.93	0.04554
hsa-miR-31-5p	5.58	0.04563
hsa-miR-425-5p	4.97	0.04713
hsa-miR-30a-3p	2.08	0.04768
hsa-miR-509-5p	1.87	0.04870
hsa-miR-130b-3p	4.42	0.04879
hsa-miR-30d-5p	2.86	0.04933
hsa-miR-4286	76.4	0.04982
hsa-miR-1915-3p	6.88	0.04983
hsa-miR-2682-5p	−1.47	0.05006
hsa-miR-130a-3p	5.11	0.05084
hsa-miR-524-3p	1.46	0.05111
hsa-miR-1234-3p	1.74	0.05119
hsa-miR-296-5p	7.06	0.05225
**hsa-miR-10a-5p**	4.83	0.05258
hsa-miR-15b-5p	8.15	0.05283
hsa-miR-876+2:77-3p	1.63	0.05330

**Table 2 ijms-25-03793-t002:** Differentially expressed miRNAs between cisplatin-resistant and cisplatin-sensitive HGSOC cell lines.

miRNA	OV-90cis vs. OV-90			OVCAR3CIS vs. OVCAR3		
	Fold Change	Behavior	*p*-Value	Fold Change	Behavior	*p*-Value
hsa-miR-200c-3p	14.0	+	0.01608	39.2	+	0.00427
hsa-miR-132-3p	67.6	+	0.01120	5.75	+	0.01266
hsa-miR-218-5p	28.7	+	0.00014	5.14	+	0.01897
hsa-miR-9-5p	27.7	+	0.00281	13.9	+	0.00289
hsa-miR-335-5p	−4.52	−	0.01551	−61.4	−	0.00318
hsa-let7b-5p	2.28	+	0.01706	7.45	+	0.00387
**hsa-miR-10a-5p**	3.08	+	0.00360	13.0	+	0.00498
**hsa-miR-141-3p**	7.51	+	0.01200	12.6	+	0.00570
**hsa-miR-296-5p**	12.5	+	0.00772	20.8	+	0.00892
hsa-miR-125a-5p	−2.45	−	0.02778	−1.73	−	0.03998
hsa-miR-15a-5p	−2.61	−	0.00682	−2.09	−	0.01410
hsa-miR-363-3p	3.15	+	0.00154	2.38	+	0.02799
hsa-miR-183-5p	3.07	+	0.00478	3.96	+	0.00300
hsa-miR-126-3p	−6.14	−	0.01289	−2.04	−	0.00524
hsa-miR-222-3p	−7.58	−	0.00417	−4.33	−	0.01030
hsa-miR-181a-5p	−11.2	−	0.00105	−2.46	−	0.01301
hsa-miR-221-3p	−7.83	−	0.00298	−10.6	−	0.00036
hsa-miR-301a-3p	−3.21	−	0.00849	−2.12	−	0.02403
hsa-miR-96-5p	2.96	+	0.01047	3.58	+	0.03423
hsa-miR-548b-3p	3.54	+	0.02254	5.78	+	0.00339
hsa-miR-30e-5p	−1.63	−	0.02687	−1.95	−	0.02206
hsa-miR-26b-5p	−2.16	−	0.04543	−2.33	−	0.03164
hsa-miR-18b-5p	4.80	+	0.01187	1.72	+	0.00754
hsa-miR-203a-3p	3.39	+	0.00476	3.80	+	0.01210
hsa-miR-182-5p	2.88	+	0.02711	5.24	+	0.03883
hsa-miR-299-3p	−1.50	−	0.02384	−1.96	−	0.04486
hsa-miR-598-3p	2.09	+	0.00867	1.70	+	0.01196
hsa-miR-92a.1-5p	−1.74	−	0.02790	−2.40	−	0.03960
**hsa-miR-1206**	2.07	+	0.03833	2.88	+	0.04735

**Table 3 ijms-25-03793-t003:** List of the nine deregulated miRNAs with their reported expression patterns and biological roles.

miRNA	Expression/Biological Role	References
miR-10a-5p	Increased levels correlated with malignant granulosa cell tumors in ovarian cancer. Knockout reduced aggressive phenotype by regulating PTEN, Akt, and Wnt pathways in granulosa cell tumor, a type of ovarian cancer. Increased in most cancers but decreased in ovarian cancer.	[8,9]
miR-96-5p	Elevated levels in OC tissues and cell lines correlate with increased proliferation and migration. Increased levels associated with reduced overall survival (OS) in serous ovarian carcinoma (SOC). Increased levels reported in both primary and metastatic SOC cases.	[10,11]
miR-141-3p	Elevated levels of miR-141 are common in OC tissues compared to noncancerous tissue. Knockdown decreased proliferation, migration, invasion, and tumor progression in OC cells. Implicated in angiogenesis, drug resistance, and metastatic colonization in OC. Paradoxically, high levels inhibited epithelial to mesenchymal transition (EMT), leading to decreased invasion and migration in the SKOV3 cells.	[12,13,14,15,16]
miR-182-5p	Overexpressed in HGSOC and clear cell ovarian carcinoma (CCC) when compared to ovarian surface epithelium (OSE). Elevated levels are detected in serous tubal intraepithelial carcinoma (STIC), precursor lesions associated with HGSOC. Elevated levels linked to increased tumor transformation, enhanced invasiveness, and metastasis in vitro and in vivo.	[17,18]
miR-183-5p	Overexpressed in OC tumor samples, with higher expression in patients at advanced clinical stages. Proposed as a potential biomarker for epithelial ovarian cancer (EOC). Inhibition associated with reduced cell viability and invasion, along with the induction of apoptosis.	[19,20]
miR-200c-3p	Overexpressed in OC tumors. Overexpression promotes metastasis and invasion by regulating EMT. Proposed as a biomarker for both HGSOC and non-HGSOC as it was detected in serum exosomes. In most studies, its overexpression is linked to poor prognosis, although inconsistent results have been reported. Upregulation increases the sensitivity of OC cells to taxanes, but contradictory results are reported with cisplatin.	[21,22,23]
miR-203a	Contrasting results regarding its expression in OC. Reduced levels observed in OC tissues compared to adjacent normal tissues. Elevated expression decreased cell migration and mouse survival but promoted a metastatic phenotype in vivo. Increased levels decreased cell proliferation, migration, and invasion, while enhancing apoptosis of OC cells via the Akt/GSK-3β/Snail pathway.	[24,25]
miR-296-5p	High levels in EOC cells reduced EMT-related proteins. Increased levels were associated with tumorigenicity and increased tumor growth. Elevated levels enhanced proliferation, migration, invasion, and drug resistance in OC cells by targeting PTEN.	[26,27,28,29]
miR-1206	Elevated levels in lung cancer tissues correlated with increased cell proliferation, blood vessel formation, migration, and invasion, while reduced apoptosis. Proposed as a potential biomarker for lymphoblastic leukemia.	[30,31]

**Table 4 ijms-25-03793-t004:** List of candidate genes regulated by four, three, and two (interactors) of nine miRNAs according to the miRTargetLink program.

Gene Symbol	Number of Interactions	miRNA 1	miRNA 2	miRNA 3	miRNA 4
PTEN	4	hsa-miR-200c-3p	hsa-miR-141-3p	hsa-miR-182-5p	hsa-miR-10a-5p
ZEB1	4	hsa-miR-200c-3p	hsa-miR-141-3p	hsa-miR-96-5p	hsa-miR-183-5p
FOXO1	4	hsa-miR-200c-3p	hsa-miR-182-5p	hsa-miR-96-5p	hsa-miR-183-5p
SNAI1	4	hsa-miR-203a-3p	hsa-miR-182-5p	hsa-miR-96-5p	hsa-miR-183-5p
ZEB2	3	hsa-miR-200c-3p	hsa-miR-141-3p	hsa-miR-203a-3p	
E2F3	3	hsa-miR-200c-3p	hsa-miR-141-3p	hsa-miR-203a-3p	
VEGFA	3	hsa-miR-200c-3p	hsa-miR-203a-3p	hsa-miR-296-5p	
BDNF	3	hsa-miR-10a-5p	hsa-miR-182-5p	hsa-miR-96-5p	
BM1	3	hsa-miR-200c-3p	hsa-miR-203a-3p	hsa-miR-183-5p	
RECK	3	hsa-miR-96-5p	hsa-miR-182-5p	hsa-miR-183-5p	
GSK3B	3	hsa-miR-96-5p	hsa-miR-182-5p	hsa-miR-183-5p	
SMAD4	3	hsa-miR-203a-3p	hsa-miR-182-5p	hsa-miR-183-5p	
TP53	2	hsa-miR-182-5p	hsa-miR-96-5p		

**Table 5 ijms-25-03793-t005:** List of candidate genes regulated by five or four miRNAs according to the Mienturnet program.

Gene Symbol	Number of Interactions	miRNA 1	miRNA 2	miRNA 3	miRNA 4	miRNA 5
ZEB1	5	hsa-miR-141-3p	hsa-miR-200c-3p	hsa-miR-183-5p	hsa-miR-96-5p	hsa-miR-203a-3p
IGF1R	5	hsa-miR-183-5p	hsa-miR-182-5p	hsa-miR-96-5p	hsa-miR-141-3p	hsa-miR-203a-3p
PTEN	4	hsa-miR-141-3p	hsa-miR-182-5p	hsa-miR-10a-5p	hsa-miR-200c-3p	
MALT1	4	hsa-miR-200c-3p	hsa-miR-141-3p	hsa-miR-96-5p	hsa-miR-182-5p	
TP53INP1	4	hsa-miR-182-5p	hsa-miR-200c-3p	hsa-miR-1206	hsa-miR-96-5p	
E2F3	4	hsa-miR-203a-3p	hsa-miR-141-3p	hsa-miR-200c-3p	hsa-miR-1206	
FOXO1	4	hsa-miR-182-5p	hsa-miR-96-5p	hsa-miR-183-5p	hsa-miR-200c-3p	
SNAI1	4	hsa-miR-182-5p	hsa-miR-203a-3p	hsa-miR-183-5p	hsa-miR-96-5p	
GSK3B	4	hsa-miR-183-5p	hsa-miR-96-5p	hsa-miR-182-5p	hsa-miR-203a-3p	
TRIP71	4	hsa-miR-182-5p	hsa-miR-96-5p	hsa-miR-183-5p	hsa-miR-203a-3p	
GATA6	4	hsa-miR-183-5p	hsa-miR-141-3p	hsa-miR-200c-3p	hsa-miR-203a-3p	
NUFIP2	4	hsa-miR-183-5p	hsa-miR-182-5p	hsa-miR-203a-3p	hsa-miR-200c-3p	
XIAP	4	hsa-miR-200c-3p	hsa-miR-96-5p	hsa-miR-183-5p	hsa-miR-10a-5p	

## Data Availability

All data generated or analyzed during this study are included in this published article (and its Appendix A).

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
