# Peer review of "MicroRNA Expression Profiles in Human Samples and Cell Lines Revealed Nine miRNAs Associated with Cisplatin Resistance in High-Grade Serous Ovarian Cancer"

_ijms, 2024, doi:10.3390/ijms25073793_

Round 1

Reviewer 1 Report

Comments and Suggestions for Authors

Dear authors, work devoted to a thorough analysis of the RNA profile of ovarian cancer samples and corresponding cell lines, with an emphasis on sensitivity to cisplatin, was identified nine key players - nine microRNAs, the levels of which are significantly increased in ovarian cancer.

There are several small comments about the work.

Since the Materials and Methods section is placed at the end of the text, many abbreviations introduced there are not deciphered in the results, which are located earlier in the text.

At the end of the introduction, it is said that the discovered microRNAs can be used as targets for the therapy of ovarian cancer, could you decipher exactly how this is possible?

Questions for Materials and Methods:

line 432 Was informed consent obtained from each patient? Was the study carried out considering the Helsinki Declaration? The number and date of the meeting of the institute's ethics committee that approves the study must be provided.

Line 436 It is necessary to add a description of the process of tissue fixation and their further processing. And give the Giemsa staining protocol.

Line 489 Were specific RNAs used or was it just one type of non-specific RNA?

Please check that superscript font is used wherever necessary.

In Figure 1 b in the photographs shown, the white balance is not set, and the brightness is not equalized. It is necessary to indicate where the Matrigel is and where the cells are. Do Matrigel and cells differ in staining?

Reviewer 2 Report

Comments and Suggestions for Authors

 MicroRNA expression profiles in human samples and cell lines revealed nine miRNAs associated with cisplatin resistance in  high-grade serous ovarian cancer, identifies 9 miRNAs associated with HGSOC.  The initial screening process uses 4 cell lines, 2 with cisplatin resistance.  From this series of screening processes winnowed  the group of candidates to 9.  Their possible link to HGSOC is further explored with miRTarget and miRTarbase link to pin down associations with specific genes.  This is validated using OMIs to impact presumptive target genes in cell lines and to alter cell clonicity and invasion.

The findings seem solid enough, with the exception that we don’t see the direct impact of OMIs on any of the miRNAs.  Can this be remedied?  My other main comments regard presentation style. In Fig 4., rearrange the panels so that the 4 K-M plots with significant p-values are in the first 4 slots and then arrange the remainder according to ranked p-value.

Highlight (maybe using bold type) these 4 miRNAs in the Tables. 

Likewise in the Discussion section, discuss these four first, then the others.

Comments on the Quality of English Language

OK

Round 2

Reviewer 2 Report

Comments and Suggestions for Authors

The paper is acceptable.  Consider adding the first two sentences of your reply to me, " We performed the in vitro experiments at least three times------" into the text of the M&M section.

Comments on the Quality of English Language

-